# Effect of a Vegan Diet on Alzheimer’s Disease

**DOI:** 10.3390/ijms232314924

**Published:** 2022-11-29

**Authors:** Alzbeta Katonova, Katerina Sheardova, Jana Amlerova, Francesco Angelucci, Jakub Hort

**Affiliations:** 1Memory Clinic, Department of Neurology, 2nd Faculty of Medicine, Charles University and Motol University Hospital, 150 06 Prague, Czech Republic; 2International Clinical Research Centre, St. Anne’s University Hospital, 602 00 Brno, Czech Republic

**Keywords:** vegan diet, Alzheimer’s disease, cognition

## Abstract

There is evidence indicating that a vegan diet could be beneficial in the prevention of neurodegenerative disorders, including Alzheimer’s disease (AD). The purpose of this review is to summarize the current knowledge on the positive and negative aspects of a vegan diet regarding the risk of AD. Regarding AD prevention, a vegan diet includes low levels of saturated fats and cholesterol, contributing to a healthy blood lipid profile. Furthermore, it is rich in phytonutrients, such as vitamins, antioxidants, and dietary fiber, that may help prevent cognitive decline. Moreover, a vegan diet contributes to the assumption of quercetin, a natural inhibitor of monoamine oxidase (MAO), which can contribute to maintaining mental health and reducing AD risk. Nonetheless, the data available do not allow an assessment of whether strict veganism is beneficial for AD prevention compared with vegetarianism or other diets. A vegan diet lacks specific vitamins and micronutrients and may result in nutritional deficiencies. Vegans not supplementing micronutrients are more prone to vitamin B12, vitamin D, and DHA deficiencies, which have been linked to AD. Thus, an evaluation of the net effect of a vegan diet on AD prevention and/or progression should be ascertained by taking into account all the positive and negative effects described here.

## 1. Introduction

In recent years, a vegan diet has grown in popularity worldwide. According to a survey conducted in 2021, vegans represent about 2–3% of the population in European countries [1]. The complex effect of a vegan diet on mortality, health, and environmental outcomes was also reflected by the recommendation for a sustainable diet strategy based on the survey from 150 countries worldwide, where a vegan diet turned out to be the most effective for the followed parameters as compared to vegetarian, pescatarian, and flexitarian diets [2]. The main reasons why people seek diets without meat and animal products can be ideological, religious, or medical [3,4]. Among these reasons, there is some evidence indicating that a vegan diet could be beneficial in preventing neurodegenerative disorders, such as Alzheimer’s disease (AD).

Neurodegenerative disorders are also on the rise worldwide. It was estimated that more than 50 million people worldwide lived with dementia in 2019, which is expected to triple in 30 years, reaching 152 million in 2050 [5]. AD is the leading cause of dementia, and its prevalence is growing rapidly, making it a major public health issue. The onset of AD is predicted by several risk factors, which are both genetic and modifiable (Table 1). Non-modifiable risk factors include advanced age [6,7,8], gender [7,8], a family history of dementia, and genetic susceptibility [6]. AD is also associated with modifiable risk factors, such as depression [9,10,11], hypertension [12,13,14], type 2 diabetes [15,16,17], obesity [18,19,20], physical inactivity [21,22], low education [6,7,23], and unhealthy diet [24,25]. Even though there is currently no cure for AD, people can reduce their risk by addressing the modifiable risk factors. One of the key lifestyle factors that can be modified to prevent AD is diet. The purpose of the present review is to summarize the current knowledge on the positive and the negative aspects of a vegan diet in the prevention and during the course of AD.

## 2. Vegan Diet and Brain Function

Nutrition plays a crucial role in maintaining proper brain function as we age. Researchers have been increasingly studying the role of dietary and lifestyle factors, such as plant-based diets, in AD [25]. Brain health was found to be improved by diets such as the Mediterranean, Mediterranean-DASH Intervention for Neurodegenerative Delay (MIND), or Dietary Approaches to Stop Hypertension (DASH). Adhering to the Mediterranean, DASH, or MIND diet can decrease cognitive decline and AD risk [26,27,28]. What these diets have in common is that they limit sugar and saturated fat intake and recommend eating a high percentage of fruits, vegetables, whole grains, and nuts and consuming minimal amounts of red or processed meat. Several medical organizations have recommended a plant-based diet to optimize cognitive health and potentially prevent dementia [29,30,31]. To maintain cognitive health and prevent cognitive aging, consuming a plant-based diet can be a low-risk and beneficial lifestyle change.

However, plant-based and vegan diets are not synonymous. A vegan diet tends to be plant-based, but plant-based diets are not vegan by definition. In a plant-based diet, plants are the primary component, but animal products are also included in small amounts. A whole-food, plant-based diet is centered on whole, unrefined plant foods and minimizes highly refined foods, such as bleached flour, refined sugar, oil, and processed packaged foods [32]. A vegan diet entirely abstains from animal products and is a stricter version of vegetarianism. In addition to cutting out meat, vegans eliminate everything made or derived from an animal, including dairy, eggs, and honey. Additionally, vegans refrain from using animal products in other areas of their lives, including beauty products, footwear, and clothing [4].

The effects of a plant-based diet on brain health and cognition are well documented. A vegan diet seems effective for various outcomes, ranging from weight loss to cardiometabolic health [33] to reduced cancer incidence [34]. Still, its effects on the brain are not well ascertained. Does strict veganism confer any more protective benefits for the brain than vegetarianism or healthy eating? The results of vegetarian diets regarding health outcomes are not necessarily applicable to a vegan diet because a vegan diet is stricter and contains only plant-based foods. A rapid increase in veganism has necessitated a better scientific understanding of the effects of a vegan diet on human health, particularly relating to the brain and cognitive functions.

## 3. Possible Beneficial Effects of a Vegan Diet on the Brain and the Risk of AD

The quality and quantity of nutrients vary according to the diet one follows. Compared to omnivorous diets, a vegan diet is typically richer in fiber; polyunsaturated fatty acids (PUFA); vitamins A, B1, B6, C, and E; folate; magnesium; iron; and copper [35,36,37]. All these nutrients can have an effect on AD pathophysiology. A vegan diet could therefore aid in either the primary or secondary prevention of AD. In this section, we will analyze each of the diet components and specify whether they are useful for the primary or secondary prevention of AD.

### 3.1. Fruits and Vegetables

Vegetables, fruits, grains, legumes, nuts, and seeds constitute the bulk of a vegan diet. Several meta-analyses have found that the increased consumption of fruits and vegetables can reduce dementia risk and slow down cognitive decline in older adults [38,39,40]. Conversely, a low vegetable intake is associated with poorer cognition in AD dementia [41]. A high intake of fruits and vegetables could therefore act as secondary prevention in AD. Furthermore, the phytochemicals, vitamins, minerals, and fiber found in fruits and vegetables have well-established anti-inflammatory and antioxidant properties, which may protect the brain by reducing the pathological processes associated with aging and dementia [42] (Figure 1).

### 3.2. Reduction in Inflammation

Inflammation plays an important role in the development of AD. Inflammatory cascades may contribute to AD pathogenesis when the amyloid beta (Aβ) levels are continuously high, mobilizing the innate immune system through microglia activation [43]. Patients with AD often present with high levels of inflammatory markers, and these markers are linked to cognitive decline as well [44,45].

Part of the protective mechanisms of a vegan diet could be attributed to its beneficial effect on the reduction of inflammatory markers (Figure 1), thus acting as secondary prevention in AD. It appears that meat-based dietary patterns are positively correlated with biomarkers of low-grade inflammation, while vegetable- and fruit-based diets are inversely correlated [46]. However, studies providing data on inflammatory biomarkers in vegans are sparse and inconsistent. In a cross-sectional study of 36 vegans and 36 omnivores, Menzel et al. found no significant differences in any of the seven investigated inflammatory biomarkers (high-sensitivity C-reactive protein (hsCRP), adiponectin, ICAM-1, IL-18, IL-1 RA, omentin-1, and resistin). The participants that adhered to the vegan diet for over 4.8 years were more likely to have lower hsCRP levels compared to those adhering to a vegan diet for less than 4.8 years [47], suggesting the length of the diet may be an essential factor in reducing systemic inflammation. Šebeková et al. also found that plasma CRP levels did not differ significantly between vegans and omnivores [48]. In contrast, Franco de Moreaes et al. detected lower values of inflammatory markers, CRP, and the TNF-α/IL-10 ratio in strict vegetarians (defined as consuming animal products less than once a month) compared to vegetarians and omnivores [49]. Suttlife et al. found reductions in the circulating CRP after a three-week lifestyle intervention that included a vegan diet [50]. Because the study included only overweight and obese participants, the decline in the CRP level between the baseline and follow-up could have been a result of various factors. The reduction in the body mass index was probably the most influential because overweight and obesity are associated with high levels of inflammation [51]. Researchers found that participants following a vegan diet prior to the intervention had the most favorable CRP profiles compared to vegetarians and omnivores [50], which may hint at the fact that a vegan diet may help to reduce inflammatory processes. Finally, a recent meta-analysis concluded that vegans have lower levels of CRP than omnivores [52].

Despite these data, the evidence regarding the effect of a vegan diet on inflammatory biomarkers is still limited. The majority of studies examining inflammatory biomarkers in vegans have small sample sizes and are cross-sectional, so causality cannot be determined. So far, only one vegan intervention study has been conducted, and because the participants were overweight, generalizing to other populations was not possible [50]. Further studies are needed to prove that a vegan diet may help prevent or counteract inflammatory states and subsequently aid in lowering the risk of AD.

### 3.3. Modifiable Risk Factors for AD

Other than genetic factors, hypertension [12,13,14], diabetes [15,16,17], obesity [18,19,20], and midlife elevated blood lipids [53,54,55,56] all increase the AD dementia risk. All these AD risk factors can be modified through a vegan diet; therefore, a vegan diet can also aid in the primary prevention of AD. As a whole, a vegan diet can indirectly improve cognition by maintaining a healthy body weight and by reducing cardiovascular risk factors, such as cholesterol [34], blood glucose [57], and blood pressure [58]. Furthermore, a vegan diet might be a valuable tool for preventing diabetes [59]. Vegan diets have a lower energy content, making the people adopting them more likely to have a healthier body mass index (BMI) and lower obesity rates than those following other diets [34,35]. Low BMIs, maintained on the vegan diet, likely contribute to improved lipid profiles, glycemic results, and insulin sensitivity.

A vegan diet influences nutrient intake in several ways, which may ultimately affect insulin sensitivity. A vegan diet increases the intake of protective nutrients, such as polyphenols [60]. Dietary polyphenols inhibit glucose absorption in the intestine, stimulate insulin secretion, and enhance insulin-dependent glucose uptake [61]. The glucose-lowering effect of a vegan diet may also be attributed to the higher fiber content. Soluble dietary fiber improves glycemic control by delaying gastric emptying and the consequent slower glucose absorption and uptake [59]. Additionally, both soluble and insoluble fiber consumption can result in improved glycemic control by increasing insulin sensitivity [62]. A low-fat vegan dietary intervention leads to lower intramyocellular and hepatocellular lipid storage and thus increased insulin sensitivity [63].

### 3.4. GI Tract

A growing body of research shows that the gut microbiome plays a crucial role in AD pathogenesis [64,65,66]. The gut microbiome of AD patients is compositionally different, and it has a decreased diversity compared with those of cognitively unimpaired people [67]. Interactions between the intestine and the brain are mediated by the nervous system or by chemical substances crossing the blood–brain barrier [68]. A dysbiotic gut microbiome may contribute to the progression and exacerbation of the disease, possibly by promoting immune activation, systemic inflammation, Aβ aggregation, and insulin resistance in the periphery and the brain [66]. A vegan diet could act as a secondary prevention measure in AD by potentially establishing a healthier gut microbiota.

The ability to target the gut microbiota and restore its composition through food-based therapy may provide new preventive and therapeutic options for AD. Diet is among the key factors affecting the gut microbiota ecosystem [69]. A vegan diet differs from an omnivorous diet in terms of its macronutrient composition. The protein, total fat, and saturated fat intake is lower, while the carbohydrate and fiber intake is higher on a vegan diet [35,70]. The macronutrient balance alters the composition of the gut microbiota and, in turn, affects the production of metabolites that may have a positive or negative effect on health. A vegan diet seems to promote a more diverse gut microbiome and a more even distribution of microbial species [71]. The diversity and stability of the vegan gut microbiota are primarily attributed to a higher intake of complex carbohydrates, fiber, and polyphenols [35,72]. However, researchers have reported mixed results about how exactly a vegan diet affects the microbiome and its diversity. A short-term, four-week vegan diet intervention in omnivorous participants that were randomized to a vegan or an omnivorous diet led to no remarkable changes in their gut microbiota [73]. No significant difference between the alpha diversity of the subjects on the vegan or omnivore diet was observed; there were, however, changes in the abundance of the genera Coprococcus, Roseburia, and Blautia after the trial, but most of them were only detectable in a few of the samples [73]. Coprococcus, which was enriched in a vegan diet and depleted in an omnivore diet [73], was previously reported to be depleted both in the gut microbiota of 3xTg-AD mice [74] and in the fecal microbiota of AD patients and positively correlated with Mini Mental State Exam (MMSE) scores [75]. Prochazkova et al. also found only modest differences in the microbiome composition associated with a long-term vegan vs. omnivore diet [76].

A recent systematic review of cross-sectional studies concluded that most studies report an increased abundance of Bacteroidetes on the phylum level and a higher abundance of Prevotella on the genus level in vegans compared to omnivores [77]. Prevotella is one of the fiber-utilizing bacterial species that ferments fiber into short-chain fatty acids (SCFAs). SCFAs are among the most abundant metabolites of the gut microbiota, and they play a crucial role in mediating gut–brain interactions. Their presence has been implicated in the occurrence and development of AD, and they play an important role in the disease process by regulating the synaptic plasticity, reducing the Aβ and tau pathology, and neuroinflammation [78]. In fecal samples, SCFAs decrease progressively from amnestic mild cognitive decline (MCI) to AD [79]. A vegan diet should in theory result in higher concentrations of SCFAs due to the increased intake of fiber, but the results of the studies are not unanimous. While Prochazkova et al. found a higher concentration of SCFAs in vegan fecal and serum metabolome [76], Trefflich et al., Wu et al., and Reiss et al. observed no significant differences in the fecal SCFA concentrations between vegans and omnivores [80,81,82]. While De Filippis et al. found that a vegan diet produced greater quantities of SCFAs, they also found a positive correlation between the SCFAs levels and adherence to a Mediterranean diet also in omnivores, irrespective of the diet type [83].

The vegan gut microbiota may also provide health benefits by reducing inflammation because it has been found to contain fewer pathobionts associated with low-grade inflammation [77]. Cattaneo et al. found that Aβ-positive AD patients have more pro-inflammatory bacteria in the gut than healthy controls and Aβ-negative patients. The inflammatory state, cognitive impairment, and Aβ presence in the brain were all positively related to the presence of pro-inflammatory microorganisms in the gut [84].

There is still some uncertainty about whether or not it leads to a more protective, healthier gut microbiota [77]. It seems that a vegan diet could be beneficial for the gut microbiota, although individual responses vary [73]. Despite the similarity between the gut microbiota of vegans and omnivores, their metabolomic profiles are quite different [76,81]. Veganism reduces the abundance of potentially harmful metabolites and increases the abundance of beneficial metabolites [76]. There is a need for further research to clarify the complicated mechanisms and interactions between the vegan diet, gut microbiota, and the subsequent effect of the diet on the pathophysiology and development of AD. Diet diversity is a critical driver of microbiota stability, and as such, it might be more essential to consume a variety of plant-based foods rather than to exclude animal products [85].

### 3.5. TMAO Reduction

Trimethylamine N-Oxide (TMAO) is the product of a microbial metabolite that increases with red meat consumption and has been linked to neurological diseases [71,72,85]. It was shown that the accumulation of tau and Aβ in the brain may be enhanced by TMAO. TMAO is capable of controlling the folding and aggregation state of the Aβ and accelerating its random coil-to-β-sheet conformational change, which is essential for the formation of Aβ fibers, thus accelerating the amyloidogenic plaque formation [86]. Accordingly, in AD and MCI patients, elevated cerebrospinal fluid TMAO levels correlate positively with the biomarkers of AD pathology and neurodegeneration [87], suggesting that TMAO may contribute to AD pathology. A possible mechanism by which TMAO could contribute to the development of AD is through the exacerbation of neurodegenerative and neuroinflammatory processes. Additionally, TMAO has the potential to contribute to AD by stimulating insulin resistance and other metabolic disturbances associated with AD pathophysiology [88]. An increase in plasma TMAO has been shown to promote brain aging and cognitive impairment and to worsen AD by reducing the neurite density and increasing the synaptic damage in mice [89,90]. Conversely, reducing the TMAO levels in plasma has been shown to ameliorate cognitive decline in a mouse model of AD [91].

Vegan diets decrease the TMAO levels in plasma and urine, thus acting as a secondary prevention in AD, while diets high in animal protein have a negative effect [92]. In an interventional study investigating the impact of a vegan diet, the TMAO levels decreased after only eight weeks of consuming a vegan diet [93]. It has also been shown that following the consumption of L-carnitine, trimethylamine that is abundant in red meat, TMAO is produced in greater quantities by the intestinal microbiota of omnivores compared to vegetarians and vegans [94]. However, the plasma concentrations of TMAO between lacto-ovo-vegetarians and vegans do not seem to differ [95], suggesting that a vegetarian diet might be also powerful enough to lower TMAO levels.

### 3.6. Mental Health

Mental health could be an important factor to prevent AD insurgence. In the case of problems associated with mental health, it is possible that the risk of getting AD will increase. Factors such as chronic stress and depression can in fact increase the risk of developing AD [9,10,11]. In this regard, there is some light evidence that a vegan diet positively affects mental health and well-being, thus acting as primary prevention in AD.

Systemic reviews on the association between a vegan diet and depression reveal conflicting evidence, possibly due to the heterogeneity of the studies included. Thus, the pitfall of the existing studies is that the causal effect between depression and a vegan diet cannot be depicted [96].

Another meta-analysis found an association between a vegan diet and lower scores of anxiety [97]. The only intervention study with a raw vegan diet reported improvement in the overall QOL by 11.5% (*p* = 0.001), a decrease in anxiety by 18.6% (*p* = 0.009), and perceived stress by 16.4% (*p* < 0.001) after 12 weeks [98].

Possible mechanisms for the effects of a vegan diet in reducing depression and/or anxiety could be related to the action of quercetin, which is found only in plant foods [99]. Quercetin acts as an inhibitor of monoamine oxidase (MAO), an enzyme that breaks down the neurotransmitters regulating mood, such as serotonin, dopamine, and norepinephrine [100]. Thus, by acting as a sort of natural antidepressant, quercetin can increase the amount of serotonin, dopamine, and norepinephrine in the brain [101], an effect that can mitigate anxiety and depression. In addition to its antidepressant properties, it was shown that quercetin exerts a neuroprotective effect in AD animal models. Mice treated with quercetin show a significant improvement in cognitive performance [102]. As a neuroprotective agent, quercetin inhibits Aβ aggregation and tauopathy [103], reduces oxidative stress and inflammation [104], and stimulates neurogenesis and neuronal plasticity by increasing the BNDF/TrkB signaling [105].

Conversely, arachidonic acid, which is found mostly in animal food, increases inflammation in the body [106], which is subsequently associated with feelings of anxiety, stress, and depression. Therefore, a diet that avoids arachidonic acid and increases quercetin could have a beneficial effect on one’s mood. Nonetheless, longitudinal interventional studies for confirming this causal effect are lacking.

## 4. Possible Detrimental Effects of a Vegan Diet on the Brain and the Risk of AD

A vegan diet is restrictive and has fewer food options, increasing the risk of various nutrient deficiencies (Table 2). Regarding the micronutrient supply, two recent systematic reviews have found significant differences between the nutrient intake of meat-eaters and vegans. Veganism was associated with a low intake of vitamins B2, B3, B12, and D; iodine; calcium; selenium; and zinc [35,36]. 

### 4.1. Vitamin B12 Deficiency

Vitamin B12 (cobalamin) is an essential vitamin that plays a vital role in the health of the brain and nervous system. The central nervous system requires vitamin B12 for the proper development and initial myelination as well as for the maintenance of the myelin sheath [110]. Upon absorption, vitamin B12 acts as a cofactor for methionine synthase, an enzyme responsible for the synthesis of methionine from homocysteine. Consequently, a vitamin B12 deficiency may result in a decreased synthesis of methionine and S-adenosylmethionine, adversely affecting methylation reactions, which play an essential role in the metabolism of myelin sheath components [111]. Vitamin B12 also acts as a coenzyme for methylmalonyl-CoA mutase to convert methylmalonyl-CoA into succinyl-CoA. A failure to perform this reaction results in the conversion of methylmalonyl-CoA into methylmalonic acid, which destabilizes the myelin sheath [111]. Thus, a low B12 status can, among other neurological symptoms, lead to progressive axonal demyelination and an increased rate of cerebral atrophy in the elderly [110,112,113]. An elevated homocysteine concentration is considered a sensitive metabolic marker for a vitamin B12 deficiency [114] and is closely linked to an increased risk of AD [115]. Clinical trials have found that lowering homocysteine levels with B vitamins treatment slows down the rate of brain atrophy and cognitive decline in people with MCI with high homocysteine levels at baseline [116,117,118]. In a vegan diet, a deficiency in vitamin B12 is likely because animal food is the primary source of this nutrient (Figure 2). Although some plant-based foods contain amounts of B12, such as soy-yogurt, mushrooms, and seaweed, these tend to be less efficient than animal foods as they contain significant quantities of vitamin B12 analogues that are biologically inactive in humans [119,120].

Therefore, it is not surprising that the intake of vitamin B12 is significantly lower in vegan diets than that of other dietary types [35,36]. An unsupplemented vegan diet will inevitably result in a B12 deficiency. In a five-year prospective study, transitioning from an omnivore to a vegan diet was linked to vitamin B-12 deficiency [107]. Nonetheless, only the participants that did not take any supplements became deficient. In contrast, the group that consumed food fortified with B12 did not exhibit any signs of deficiency [107], confirming that foods fortified with B12 and B12 supplements are the only reliable vegan sources of B12.

Even a short four-week vegan diet intervention decreased the bioactive form of vitamin B12 holotranscobalamin in plasma [121]. A low intake of dietary B12 does not necessarily result in a deficiency. According to cross-sectional studies comparing the micronutrient status of vegans and omnivorous diets, vegans consume negligible amounts of dietary vitamin B12. Still, their vitamin B12 status is mainly normal, and a deficiency in this particular vitamin is rare [37,108]. Vegans who regularly supplement with cobalamin have similar serum cobalamin levels to nonvegans [122]. Most vegans nowadays are aware of the risk of a vitamin B12 deficiency associated with the diet, and vitamin B12 is one of the most commonly taken supplements [37]. In general, vegans do not have to be worried about a deficiency leading to a high homocysteine level if they supplement regularly.

### 4.2. Vitamin D Deficiency

A vitamin D deficiency is not a unique feature of the vegan diet [123], but because plant foods that naturally provide vitamin D are few, a vegan diet is generally associated with a lower intake of vitamin D when compared to other diet types [35]. In the Nutritional Evaluation Study, Dawczynski and coworkers found that while the dietary intake of vitamin D was below the recommendations in all diet groups (omnivore, flexitarian, vegetarian, and vegan), it was by far the lowest in vegans [70]. Schupbach et al. observed the same trend in Switzerland vegans, vegetarians, and omnivores [108]. Among Danish vegans, the intake of vitamin D, including supplements, did not reach the 2012 Nordic Nutrition Recommendation of 10 μg/day [109]. The lower levels of vitamin D intake among vegans are not surprising considering that animal foods are the major dietary sources of vitamin D. Nonetheless, vitamin D can also be synthesized in the skin post-sunlight exposure, consumed in fortified foods (plant milk, tofu, and orange juice), and it can be taken as a dietary supplement. Two forms of vitamin D supplements are available: vitamin D2 (ergocalciferol) and vitamin D3 (cholecalciferol). As ergocalciferol is produced in plants and fungi, it is always suitable for vegans, while cholecalciferol can be derived from either sheep wool (a nonvegan source) or lichen (a vegan-friendly source).

Vitamin D has a well-established neuroprotective effect in the central nervous system (CNS) [124]; therefore, its deficiency may potentially play a role in the development of AD. The neuroprotective role of vitamin D in the CNS could be due to several factors, including the upregulation of neurotrophic factor production, Aβ clearance, oxidative stress mechanisms, neuronal calcium homeostasis, and immune modulation [124]. Accordingly, low vitamin D concentrations in blood have been linked to cognitive impairment and dementia in observational studies [125,126,127]. Thus, a vitamin D deficiency associated with a vegan diet could be seen as a potential increasing factor for AD development [128] (Figure 2). This assumption is supported by several data in AD patients. When absorbed, vitamin D is transported to the liver where it is converted into 25-(OH)D, which can be measured in the blood. AD patients have lower serum 25(OH)D levels than healthy controls [129] and individuals with MCI [130]. In addition, MCI patients who develop AD suffer from significantly lower 25(OH)D levels compared to MCI patients who do not progress to AD [130]. The meta-analyses of prospective cohort studies on vitamin D concentration and dementia generally support the fact that there is an association between vitamin D deficiency and a higher risk of AD [126,127,128]. A recent umbrella review of observational studies and randomized controlled trials confirmed the finding that lower concentrations of vitamin D were indeed associated with a higher risk for AD [125]. Furthermore, this association seems to be stronger in the case of a severe vitamin D deficiency (<25 ng/mL) as compared to a moderate vitamin D insufficiency (25–49.9 ng/mL), suggesting the possibility of a dose–response association between serum 25(OH)D and the risk of AD or dementia [126,128,131].

### 4.3. Omega-3 Polyunsaturated Fatty Acids Deficiency

α-linolenic (ALA), eicosapentaenoic (EPA), and docosahexaenoic acids (DHA), classified as omega-3 fatty polyunsaturated acids (n-3 PUFAs), constitute the primary source of fat in a vegan diet [35,60,132,133]. The essential fatty acid ALA is the most prevalent plant-based source. EPA and DHA, however, occur primarily in seafood and fish oil, making them likely to be of concern to vegans. EPA and DHA can be converted from ALA and thus are not considered essential. However, many factors influence the relationship between the n-3 diet intake, bioavailability, and conversion [134]. Humans generally convert ALA into DHA and EPA poorly [135]. Most studies indicate that vegans have low or no dietary intakes of DHA or EPA [36,132,134]; however, the clinical significance of this is unknown. Vegetarians do not exhibit clinical signs of a DHA deficiency [136]. Because plant-based sources of EPA and DHA are lacking, vegans must rely mainly on endogenous synthesis for long-chain omega-3 PUFAs or supplement with algae oil capsules.

The positive effects of n-3 PUFAs on AD may be due to various mechanisms—an increasing clearance of Aβ, increasing neurotrophic and neuroprotective factors, and anti-inflammatory mechanisms [137]. N-3 PUFAs are precursors of specialized pro-resolving mediators (SPMs) that act as powerful anti-inflammatory agents [138]. In AD, the ability of peripheral blood mononuclear cells to make SPMs decreases over time [139]. The levels of SPMs are reduced in the cerebrospinal fluid samples of AD patients and positively correlate with MMSE scores [140], indicating that deficient SPM production may contribute to cognitive decline in AD. Six months of n-3 PUFAs supplementation is able to prevent this reduction in SPMs production in AD patients [139]. Transthyretin is a protein that binds Aβ and inhibits its toxicity. In a 12-month long randomized controlled trial, n-3 PUFAs supplementation was found to increase plasma transthyretin in AD patients [141]. Through its ability to shift the proportion of amyloidogenic and non-amyloidogenic amyloid precursor protein processing to non-amyloidogenic processing, DHA effectively reduces the Aβ generation and release [142]. In addition, n-3 PUFAs have also been reported to stimulate the Aβ42 phagocytosis and the production of the brain-derived neurotrophic factor while simultaneously suppressing the expression of pro-inflammatory cytokines in vitro [143].

The effects of DHA on mental and cognitive function are particularly important. It has been reported that patients with AD have lower levels of DHA in the serum [144]. Among patients with AD receiving acetylcholinesterase inhibitors, lower blood n-3 PUFAs levels, particularly DHA, were associated with a higher cognitive decline risk [145]. Some studies have demonstrated a correlation between higher blood levels of DHA and a reduced AD risk [146,147], while others have not [148,149]. The response to DHA interventions can be heterogeneous, depending on both the DHA level and cognitive status at baseline. N-3 PUFAs consumption is associated with a lower risk of dementia and a reduced likelihood of cognitive decline in patients with early-stage AD; however, it does not seem to provide any cognitive benefits in patients with late-stage AD [150,151,152,153]. A three-year investigation of n-3 PUFAs supplementation did not show significant improvements in cognitive decline in older people with memory complaints overall [154], but there were significant benefits in those with a low omega-3 status at baseline [155], suggesting that individuals with a low omega-3 index, such as vegans, may especially benefit from n-3 PUFAs supplementation. However, there is some evidence that a well-planned vegan diet with sufficient sources of ALA, including walnuts, flax seeds, chia seeds, and leafy green vegetables, might be sufficient for good mental health [156]. There is some evidence that vegetarians have a lower risk of dementia [157] and no evidence about the association of a vegan diet with dementia.

The use of DHA and EPA supplements is currently recommended in vegans; however, the entitlement of this recommendation requires more research, especially concerning possible side effects because there is some evidence that higher plasma n-3 PUFAs levels are associated with an increased prostate cancer risk [158].

## 5. Conclusions and Future Directions

A growing number of people are choosing to reduce or eliminate animal products from their diet. Given that AD pathology is strongly associated with diabetes, obesity, insulin resistance, or cardiovascular diseases, preventative strategies such as nutritional interventions may help reduce the risk of AD. However, large-scale, long-term research is needed to verify this effect, and the vegan diet has not yet been studied for its long-term effects. The possible advantage of a vegan diet lowering the chances of developing AD has been studied primarily through observational studies. Because of this, it is difficult to draw definitive conclusions about whether the benefits are directly attributable to the vegan diet. Moreover, there is the possibility that the beneficial effects of a vegan diet could be counterbalanced by the lack of specific nutrients.

Vegan diets lack specific vitamins and micronutrients and may result in some nutritional deficiencies. Vegans not supplementing micronutrients are more prone to vitamin B12, vitamin D, and DHA deficiencies, which have been linked to AD. Conversely, a vegan diet includes low levels of saturated fats and cholesterol, contributing to a healthy blood lipid profile. Furthermore, it is rich in phytonutrients, such as vitamins, antioxidants, and dietary fiber, that may help prevent cognitive decline.

Do the shortcomings overshadow the positive nutritional aspects of an animal product-free lifestyle? Whether vegan or nonvegan, any dietary practice can be detrimental to health if essential nutrients are not consumed appropriately. Vegan diets are followed for a variety of reasons, and this may affect the quality of the diet, which in turn affects brain health. The availability of highly processed vegan foods has increased, making it possible to eat a vegan diet without eating much whole-plant food. To reap the benefits of veganism, consuming a balanced diet with a wide variety of foods is essential. Ultimately, vegan diets must be closely monitored and supplemented, but they can provide adequate nutrition for all stages of life when planned carefully.

The data available to date do not allow an assessment of whether strict veganism is beneficial for brain health or AD risk compared with vegetarianism or diets with occasional meat consumption. Increasing plant-based foods may positively affect cognitive health and perhaps prevent AD. However, it is difficult to find evidence of the effects of a strictly vegan diet on cognition. Plant-based diets have a more extensive body of research that indicates protective effects on brain health. Further, it is unclear if the beneficial health effects are due to the avoidance of harmful effects associated with excessive calories and meat consumption, the specific dietary nutrients and bioactive compounds found in plants, or a combination of both. Randomized controlled trials and prospective cohort studies comparing veganism to other popular diets are scarce. The evidence provided in this review indicates that the net effect of a vegan diet on AD prevention and/or progression should be ascertained by taking into account all the positive and negative effects described here. It is advised that future studies on a vegan diet in AD should include supplementation with specific micronutrients to optimize any positive effects.

## Figures and Tables

**Figure 1 ijms-23-14924-f001:**
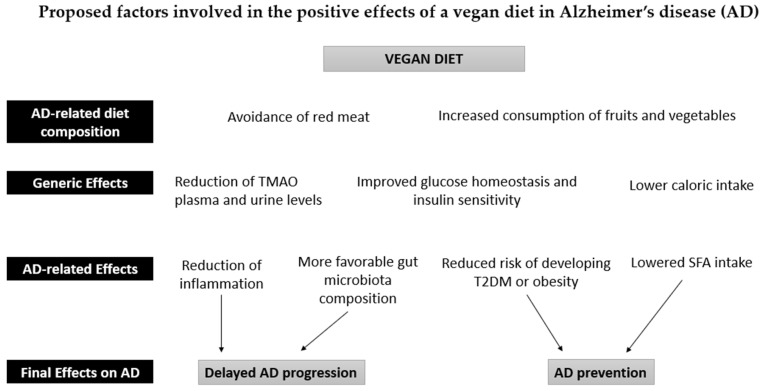
Proposed factors involved in the positive effects of a vegan diet in Alzheimer’s disease (AD). The intake of fruits and vegetables in a vegan diet is associated with a reduction in inflammatory processes and a normalization of the gut microbiota due to the high antioxidant, fiber, and polyphenol content. The absence of red meat also leads to a reduction in Trimethylamine N-Oxide (TMAO) in plasma and urine. These effects can counteract the development of AD pathology. Furthermore, the reduction in caloric intake reduces the risk of obesity and type 2 diabetes (T2DM), thus contributing to the prevention of AD, together with the reduced intake of saturated fatty acids (SFA).

**Figure 2 ijms-23-14924-f002:**
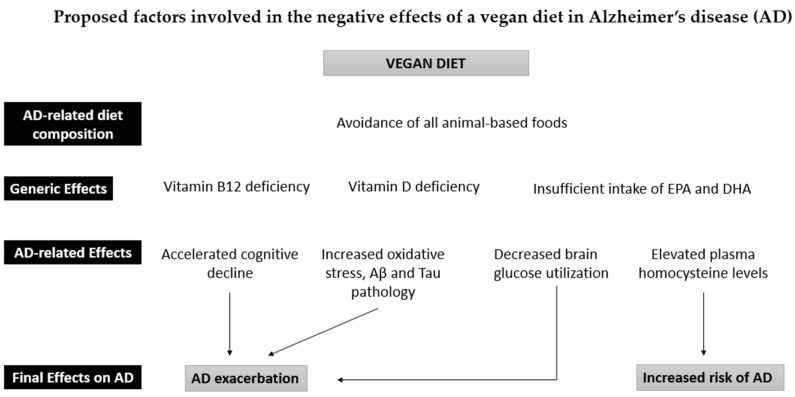
Proposed factors involved in the negative effects of a vegan diet in Alzheimer’s disease (AD). A vegan diet can easily lead to a deficiency in vitamins B12 and D if unsupplemented. Vitamin B12 deficiency accelerates cognitive decline, while a lack of vitamin D increases oxidative stress and Aβ and tau pathology. An insufficient intake of eicosapentaenoic acid (EPA) and docosahexaenoic acid (DHA), on the other hand, reduces brain glucose utilization. These factors can lead to the worsening of AD. Furthermore, these vitamin deficiencies lead to an increased risk of AD, both directly and indirectly (increase in plasma homocysteine levels).

**Table 1 ijms-23-14924-t001:** Modifiable and non-modifiable risk factors for Alzheimer’s disease.

Non-Modifiable Risk Factors	Non-Modifiable Risk Factors
Advanced age [6,7,8]	Depression [9,10,11]
Gender [7,8]	Hypertension [12,13,14]
Genetic predisposition [26]	Diabetes [15,16,17]
	Obesity [18,19,20]
	Physical inactivity [21,22]
	Low education [6,7,23]
	Unhealthy diet [24,25]

**Table 2 ijms-23-14924-t002:** Positive and negative effects of a vegan diet on Alzheimer’s disease.

Positive Effects	Negative Effects
Reduction in systemic inflammation [49,50,52,76]	Increased risk of vitamin B12 deficiency [35,36,107]
Reduced risk of developing obesity [34] and type II diabetes [59]	Increased risk of vitamin D deficiency [70,108,109]
Reduction in TMAO levels in plasma and urine [92,93]	Insufficient intake of DHA and EPA [70]
Lower caloric intake and saturated fat intake compared to other types of diets [35,70]	
Decreased pro-inflammatory bacteria in the gut [77,84]	
Increased production of beneficial bacterially produced metabolites [76,81]	

TMAO: Trimethylamine N-Oxide; DHA: docosahexaenoic acid; EPA: eicosapentaenoic acid.

## Data Availability

Not applicable.

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
