# Peer review of "Effect of a Vegan Diet on Alzheimer’s Disease"

_ijms, 2022, doi:10.3390/ijms232314924_

Round 1
Reviewer 1 Report
Dear Dr.,
Title: Effect of a Vegan Diet in Alzheimer´s Disease
Manuscript ID: ijms-2046461
Overall comments: Authors described in this manuscript: the beneficial role of vegan diet in brain health and Alzheimer´s Disease. Vegans diet not supplementing micronutrients and their role in AD prevention and/or progression remains ascertain with positive and negative effects. However, some of the contents need to improve for the better quality of this manuscript.
Specific comments:
1. The abstract is well written.
2. Trimethylamine N-Oxide (microbial metabolite) role in AD pathology regulation need to explain with additional information.
3. In section 3.6 Mental health: needs to rewrite with more clear statements and good correlation. There are multiple small paragraphs. Need to rewrite with clear statements.
4. Can be incorporating the table for modifiable and non-modifiable risk factors for AD.
5. Some of the references need to cite with more clarity and relevant to text.
6. Overall, covers the effect of vegan diet on AD prevention and progression with all positive and negative effects. However, possible overcome of vegan diet associated positive and negative effects in AD patients is missing. Hence need more information regarding these aspects to make the better outcome of this review.
*****

Author Response
Reviewer 1:
Point 1:
The abstract is well written.
Answer: Thank you. We have incorporated a sentence on Quercetin, as requested by Reviewer 2.
Point 2:
Trimethylamine N-Oxide (microbial metabolite) role in AD pathology regulation need to explain with additional information.
Answer: Additional information on Trimethylamine N-Oxide (microbial metabolite) role in AD pathology regulation has been incorporated in the revised version.
Point 3:
In section 3.6 Mental health: needs to rewrite with more clear statements and good correlation. There are multiple small paragraphs. Need to rewrite with clear statements.
Answer: Mental health section has been rewritten for clarity, as requested.
Point 4:
Can be incorporating the table for modifiable and non-modifiable risk factors for AD.
Answer: A new table for modifiable and non-modifiable risk factors for AD has been included in the revised version, as suggested.
Point 5:
Some of the references need to cite with more clarity and relevant to text.
Answer: New references have been added and cited properly.
Point 6:
Overall, covers the effect of vegan diet on AD prevention and progression with all positive and negative effects. However, possible overcome of vegan diet associated positive and negative effects in AD patients is missing. Hence need more information regarding these aspects to make the better outcome of this review.
Answer: We did our best to include all past and recent studies on the effect of vegan diet on AD prevention and progression. With the evidences provided in this review, we feel that it will be premature to reach a definitive conclusion on the effect of vegan diet and AD. On the other hand, we think that this review will help scientists to better ascertain any positive and negative effects of vegan diet on AD.
Reviewer 2 Report
Well written, although the larger part is already established knowledge. The interesting parts include TMAO and quercetin and those parts should be more extended including all recent publications, such as per example the study of Cardoza et al. (Arrona Cardoza P, Spillane MB, Morales Marroquin E. Alzheimer's disease and gut microbiota: does trimethylamine N-oxide (TMAO) play a role? Nutr Rev. 2022 Jan 10;80(2):271-281. doi: 10.1093/nutrit/nuab022. PMID: 33942080.)
Those components should be also refered to in the abstract session.
Author Response
Reviewer 2:
Point 1:
Well written, although the larger part is already established knowledge. The interesting parts include TMAO and quercetin and those parts should be more extended including all recent publications, such as per example the study of Cardoza et al. (Arrona Cardoza P, Spillane MB, Morales Marroquin E. Alzheimer's disease and gut microbiota: does trimethylamine N-oxide (TMAO) play a role? Nutr Rev. 2022 Jan 10;80(2):271-281. doi: 10.1093/nutrit/nuab022. PMID: 33942080.)
Those components should be also refered to in the abstract session.
Answer: Additional information and references on Quercetin and TMAO has now been incorporated in the revised version. Moreover, Quercetin and TMAO have been mentioned in the abstract, in accordance to the limit of word numbers allowed (200 words).
Reviewer 3 Report
the work is novel
Author Response
Reviewer 3:
Point 1:
The work is novel
Answer: Thank you very much for your positive feedback.